# Good Practices and Learning Strategies of Undergraduate University Students

**DOI:** 10.3390/ijerph17061849

**Published:** 2020-03-12

**Authors:** Margarita Gozalo, Benito León-del-Barco, Santiago Mendo-Lázaro

**Affiliations:** Department of Psychology, Faculty of Teacher Training College, University of Extremadura, 10071 Cáceres, Spain; bleon@unex.es (B.L.-d.-B.); smendo@unex.es (S.M.-L.)

**Keywords:** good practices, learning strategies, cooperation, university students

## Abstract

The educational models currently in use in higher education aim to make students active participants in their learning process, while the lecturer is seen more as a facilitator of the said process. Students’ learning strategies (superficial approach—memorizing, deep approach—looking for meaning, and achievement approach—aimed at results) and their good practices are gaining in importance and the aim of this study is to identify university students’ good practices, which are related to their learning strategies. To do so, our research covered 610 students from different science degree courses at the University of Extremadura who anonymously completed the University Students’ Good Practice Inventory (IBPEU) and the University students’ Questionnaire to Evaluate Study and Learning Processes (CEPEA). The influence of context, understood here as the center or scientific field, was discarded. The factor ‘Actively learning’ was positively associated with the deep and achievement approaches; the factor ‘Interaction with lecturers’ was positively associated with the superficial approach and negatively with the deep approach; the factor ‘Cooperative work’ was also associated with the deep approach; while the achievement approach was positively associated with the factor ‘Optimizing time’ and negatively with ‘respect for different capacities’. These associations are promising as good practices can be learnt and evaluated.

## 1. Introduction

Whenever students start to learn, there are two essential questions: Why are they doing it? And how do they do it? The answer to the first question, of course, depends on motivational factors. Answering the second involves looking at the chosen strategies and/or approaches to carry out the task: *motivation* and *strategy,* that is, the approach to learning adopted by the student [1]. A learning approach includes the student’s intentions when faced with the tasks and the processes and strategies she/he uses to carry them out [2,3,4,5].

The concept of learning strategies has been interpreted in different ways. In our case, we are not looking at them as particular resources or study techniques, but as a way of “learning to learn”, in which the students look for the meaning of what they are learning, develop their learning skills and explore their possibilities and limitations [1,5,6,7,8,9,10,11,12,13].

There are basically three different learning approaches described in the scientific literature. First, a superficial approach, the main motivation of which is to avoid failure, as well as the desire to “survive” the academic demands using the least possible effort. Thus, the learning strategies are limited to selective memorization. Secondly, an approach focused on achievement, characterized by the desire to maximize the academic qualifications using strategies that allow space and time to be organized efficiently [6]. In this approach, the strategies go beyond the mere task of learning: students organize when, where and how long they dedicate to learning, etc. [14,15]. Finally, there is the deep approach, for which the motivation is intrinsic to the task itself and is accompanied by strategies that search for meaning.

One of the aspects that have attracted most interest in research into learning approaches has been to explore which factors favor the adoption of the deep approach [16,17,18]. Many aspects of the context could interact with an individual’s personal characteristics and influence both the approach and the quality of the learning outcomes [19]. Two systematic reviews, from 2010 and 2014, provide sample information concerning the approaches, which Biggs called factors ‘that may encourage or discourage the adoption of a deep approach to learning’, of students’ choices of different learning strategies [18,20], and which he classified into a) characteristics of the student and b) contextual factors and the perception of the same. 

As for the characteristics of the students, the most studied variables have been gender, personality, study habits, and preferences concerning teaching methods and others [18]. The variable gender has given inconclusive results. In some studies, males obtained higher scores in the superficial approach [21,22], though other studies found the opposite [23,24]. There are even some studies in which females obtained higher scores in the deep approach [22,25]. A comparative study of Spain, Greece and the United Kingdom, using the R-SPQ-2F, observed differences between the sexes [26], although the results from one country to another were dissimilar. In many other studies, no gender differences were noted [27,28]. Furthermore, connections between the adoption of the deep approach and the students’ personality were found: being open to experiences, extroverted, conscientious, compassionate, empathic and trustworthy all correlate positively with the deep approach and negatively with the superficial approach. Similarly, connections were found between the superficial approach and opposite characteristics connected to neuroticism (emotional instability, anxiety and pessimism) [29,30,31].

Connections were also found between the deep approach and the time students dedicated to personal work: taking notes and consulting textbooks. The deep approach has been associated with a preference for study methods based on understanding the contents; while the learners with a superficial approach prefer methods based on the transmission of information and are more superficial and apathetic [19,32]. Similarly, the preferences of students concerning interactive teaching (such as laboratory work, reduced groups, tutorials and discussion groups) and other non-interactive modalities (such as lectures and self-study) have also been investigated [30]. The results indicated the preference of students with the deep approach for interactive methodologies and the opposite for students preferring the superficial approach. 

As for the contextual variables, the most studied has been the teaching method, its perception and the degree being studied. The teaching approaches adopted by the lecturers tend to be associated with different learning strategies. This is not due to an imposition by the lecturer, but depends on the proposed learning activities [33]. The structure of the task itself is what propitiates deep learning strategies [34]. Those lecturers who base their teaching on a mere presentation of contents, and transferring information to the student, will orient their students towards learning processes based on the superficial approach and associated with a low quality of academic achievement [33,35,36]. On the other hand, the perception of teaching characterized by support for students, feedback, freedom to learn, and clear and relevant objectives for professional practice, is associated with the deep approach [37]. Lecturers who propitiate the autonomy and independence of their students orient them towards deep learning strategies and a greater commitment to their studies [37,38].

The characteristics of the degree course can determine the type of task and the contents presented to the students; this in turn can influence the learning strategies. Several studies have observed that the superficial approach is more common in science degrees, while the deep approach is more common in humanities, which may be due to the nature of the disciplines [5,26,39].

### The Present Study

Learning centered approaches are associated with the more active and independent students; the lecturer’s role changes to become a learning facilitator, and knowledge is considered to be a tool instead of an objective [40]. Although it is thought that teaching approaches focused on the students will stimulate a greater involvement on their part in the teaching–learning processes, studies developed in the university context do not offer conclusive results about which concrete factors in this process can promote deep learning strategies in the students [10]. 

That is why, in this research, we have focused on exploring what relationships the students’ behavior and their manner of interacting with the learning contexts can have with the strategies that define their learning. To do so, we draw on the model of Chickering and Gamson [41], and all its later derivations, concerning the study of Good Practices in Higher Education. The said model is based on attitudes and behavior patterns that favor learning and the students’ theoretical and methodological development and which began in the USA in the 1980s. These authors developed a project for The American Association of Higher Education, the Education Commission of the United States and the Johnson Foundation, giving rise to the seven Principles for Good Practice in Undergraduate Education and the inventories based on these said principles [42].

In 1991, Chickering and Gamson disseminated their ideas in the book titled: ‘Applying Seven Principles for Practice in Undergraduate Education’ [43]. This text has been considered one of the best guides whereby lecturers can increase the involvement of students in their classes. Buller [44] applied these seven principles as a guide, instruction or orientation in order to discover up to what point his way of working reflected these good practices in promoting active learning and in the students’ involvement in their learning process. Some authors consider that the said principles are a reference for lecturers who wish to improve their educational strategies [44].

In order to evaluate the students’ behavior and their way of interacting with the learning contexts, we take the Inventory of Good Practices for University Students (IBPEU), originally by Pinheiro [45], as our basis. The majority of the behavior patterns evaluated using this instrument are related to intra- and inter-personal skills, many of which may be promoted and trained within the context of the university itself. On balance these resources, or strengths and weaknesses, of a psychosocial and relational nature, help us to understand why different students react in different ways to the same difficulties or challenges inherent to the academic context. 

In short, this work aims to analyze which attitudes and behavior patterns of students, associated with the Good Practices in Higher Education, predict the different learning strategies. 

## 2. Materials and Methods 

### 2.1. Participants 

The inclusion criterion for participants was to be enrolled in an undergraduate course at the University of Extremadura (Spain). The sample of participants was made up of 610 students, 42.6% (n = 260) female and 57.4% (n = 350) male, with an average age of 20.7 years (DT = 3.43; range 18–55). The students were studying degrees belonging to different scientific fields: Legal and Social Sciences, Health Sciences and Technical Education. The selection of students was done through a multistage sampling, by clusters and random selection of Degree course and year, in the Faculties of the University of Extremadura.

### 2.2. Instruments

#### 2.2.1. Questionnaire to Evaluate the Learning and Study Process of University Students

This questionnaire evaluates the level and degree of the learning approaches adopted by university students (CEPEA; Cuestionario de Evaluación de Procesos de Estudio y Aprendizaje para el Alumnado Universitario [46]) in their study process and the most relevant strategies that make up the said learning approaches. It is made up of 42 items divided into six dimensions or subscales, three of which evaluate the motivation: Superficial Motivation, Deep Motivation and Achievement Motivation; while the other three evaluate the learning strategies: Superficial Strategy, Deep Strategy and Achievement Strategy. Each of these subscales is measured through seven items. The answers are in a Likert type format with five intervals that go from 1 “Totally disagree” to 5 “Totally agree”. In this research, we have worked with the dimensions that evaluate learning strategies: the Superficial Strategy refers to those learning strategies based on the selection, memorization and reproduction of parts of the information. It is basically limited to discovering the essential parts of the subject and reproducing them through learning by rote (e.g., *“I believe complementing class notes is a waste of time, so I only seriously study what is done in class”*); the Deep Strategy refers to those strategies that search for meanings and significant learning based on reading in depth, relating new content to prior relevant knowledge (e.g., *“I try to relate what I have learnt in a subject to what I have learnt in other subjects”*); while the Achievement Strategy is based on examining in depth all the suggested reading matter and on properly organizing time and materials or resources so as to obtain good, or the best possible, qualifications (e.g., *“I summarize the suggested reading from the bibliography and include it in my notes”*). 

The indices of internal consistency (Table 1), with values between 0.711 and 0.777, show an adequate reliability of the three dimensions evaluated from the CEPEA.

#### 2.2.2. Inventory of Good Practices of the University Student

Translated to Spanish, and adapted for use with university students within the context of the European Higher Education Area (EHEA) [47,48], this questionnaire evaluates different patterns of behavior by students that can be considered adequate for performing well at university (IBPEU; Inventario de Buenas Prácticas del Alumno Universitario [47]). It is made up of 63 items divided into 9 factors or dimensions, each with 7 items. The answers are a Likert type format with 5 alternatives (Never, Rarely, Sometimes, Frequently and Always). Number 1) ‘Interaction with lecturers’, refers to students’ interest in getting to know their lecturers and looking to make contact with them inside and outside the classroom (e.g., *“I talk to my lecturers outside the classroom about the subjects, their content and other matters”*). Number 2) ‘Cooperative work with fellow students’, refers to students’ preference for learning in cooperation instead of doing so in an individualistic, competitive way; helping fellow students, sharing and debating ideas with others, stimulates deeper learning (e.g., *“Outside the classroom, I study or work in a group with other students”*). Number 3) ‘Actively learning’, refers to the students’ tendency to become involved in the learning process instead of just being a passive recipient; speaking and writing about what they are learning, relating it to past experiences and applying it to their day to day work (e.g., *“I look for experiences in my life that complement my learning in the degree subjects”*). Number 4) ‘Looking for feedback’, refers to tendency students have of being aware of their progress and receiving relevant information to improve (e.g., *“If something is not clear for me, I try to speak to the lecturers about it as soon as possible”*). Number 5) ‘Optimizing the time taken to do tasks’, reflects their intention to take advantage of the time spent learning, using the available resources, revising, planning and keeping to foreseen deadlines (e.g., *“I finish work in the specified time”*). Number 6) ‘Maintaining positive expectations’, is related to students’ intentions to improve their performance, establishing realistic but demanding goals and making an effort to keep to them (e.g., *“I try to give my best in all subjects”*). Number 7) ‘Respecting different capacities’, refers to the students’ favorable attitude towards different learning styles (e.g., *“I share information about myself and my way of learning with my fellow students”*). Number 8) ‘Managing academic challenges and opportunities’, refers to the students’ ability to reach their goals and objectives using the resources offered by the university (e.g., *“I take advantage of the learning opportunities offered by the university”*). Finally, number 9) ‘Managing personal and social resources’, refers to the skills perceived by the students to adapt to the university environment, managing their own social, emotional and motivational resources, (e.g., *“I adapt easily to new demands and academic pressures”*).

The indices of internal consistency (Table 1), with values between 0.761 and 0.929, show a good reliability of the dimensions of the IBPEU.

## 3. Results

### 3.1. Analysis of the Context Bias in the Superficial, Achievement and Deep Strategy Variables: Center Level

The students who participated in our research were from several different faculties, which could possibly influence the dependent variables studied due to their characteristics, such as the diversity of degrees, the number of students registered, or the faculty’s management modalities. The need to control this possible relationship between the students and the center in which they studied led us to apply multivariant regression models that adjusted to the nested or hierarchical data. These models presuppose that students from the same context will tend to show similar behavior patterns. 

We have subjected the data to a Random Effects Anova (Null Model) and Table 2 shows the estimates of the covariance parameters, that is, the estimates of the parameters associated to the random effects of the model.

The variance of the Centers/Faculties factor indicates how much strategies vary between faculties; while the variance of the residues indicates how much the strategies vary within each faculty. It can be seen that the Centers/Faculties factor does not have a significant influence on the dependent variables (Deep, Achievement and Superficial Strategies).

Furthermore, to calculate the existing variability between the different faculties in comparison with the existing variability between the students of the same center, we obtained the Interclass Correlation Coefficient (ICC). Values between 0 and 0.39 indicate that the subjects in the same group are as different from each other as from those in other groups; as the values obtained in the Deep (ICC = 0.20), Achievement (ICC = 0.11) and Superficial (ICC = 0.04) Strategies show, only a very small percentage of the total variability of the dependent variables corresponds to the difference between the faculty averages. In short, the center does not contribute to explaining the variability of the dependent variables. Thus, the observations are independent and the application of traditional linear models would be justified.

### 3.2. Good Practices and Learning Strategies: Regression Analysis 

In order to check whether the different dimensions of the students’ good practices are associated with the different learning strategies, controlling the gender variable creates three predictive models in steps for the three strategies: Deep, Achievement and Superficial (Table 3).

The predictive model for the Deep approach explains 29% of the variance, *F* (3, 703) = 21.593, *p* < 0.001; the predictive model for the Achievement approach explains 31% of the variance, *F*(3, 703) = 18.726, *p* < 0.001; and the predictive model for the Superficial approach explains 23% of the variance, *F*(3, 703) = 40.422, *p* < 0.001.

With respect to the Deep approach, there is a positive association with the factors ‘Cooperative work’ and ‘Actively learning’, but a negative one with the factor ‘Interaction with lecturers’. As for the Achievement approach, there is a positive association with the factors ‘Optimizing time taken to do Tasks’ and ‘Actively learning’, but a negative one with the factor ‘Respect for different capacities’. For the Superficial approach, there is a positive association with the factor ‘Interaction with lecturers’ (Table 2).

### 3.3. Interpretation of the Associations found: Classification Tree

Furthermore, in order to clarify the interpretation of the associations found in the regression analyses, classification trees were made for the Deep Strategy and Achievement Strategy variables, introducing the Good Practices of the university students as independent variables, classified by a criterion of percentiles into low (*p* ≤ 33), medium (*p* > 33 to *p* ≤ 66) and high (*p* > 66). 

The classification tree for the Deep Strategy (Figure 1) shows that the university students with high scores in ‘Actively learning’ and medium or high scores in ‘Cooperative work’ are the students who obtain the highest scores in the Deep Strategy (Node 6), while the students with low or medium scores in ‘Actively learning’ and medium or high scores in ‘Interaction with lecturers’ are the ones who obtain the lowest scores in the Deep Strategy (Node 4).

The classification tree for the Achievement Strategy (Figure 2) shows that the university students with high scores in ‘Optimizing time taken to do tasks’ and medium or low scores in ‘Actively learning’ are the ones who obtain the highest scores in the Achievement Strategy (Node 7); while the students with low or medium scores in ‘Optimizing time taken to do tasks’ and low scores in ‘Actively learning’ are the ones who obtain the lowest scores in the Achievement Strategy (Node 4). 

## 4. Discussion

This research has focused on analyzing students’ attitudes and behavior patterns associated with Good Practices in Higher Education that are related to the said students’ learning strategies. These learning strategies are not personality traits, but neither are they independent of the teaching context or the students’ preferences when faced with a task [49,50]. Within the contextual factors, it is necessary to consider a factor common to all the participants, and that is the context of the learning model focused on the students, one which is proposed in the European Higher Education Area. We have examined the differences between degree titles that represent different academic fields, observing no differences attributable to the various degree titles in the students’ learning strategies. Our results are similar to those obtained in previous studies [28,51]. 

As for our principal goal, through the regression analysis, we have been able to interpret that there is a clear relationship between the Good Practices assumed by students and the different learning strategies. What is the situation with the superficial approach? Those students who followed a Superficial Strategy are characterized by looking to ‘Interaction with lecturers’, as happens with those who have low scores in the Deep approach (Table 2). This strategy is associated with teaching styles based on the mere presentation and reproduction of the contents, where the focus of attention is centered on the figure of the lecturer and the students are seldom asked to participate [3,4,19]. This teaching model does not favor student autonomy, it simply transfers information to the students through the presentation of the contents [33,35,36]. The Superficial approach has not been related to the factor ‘Optimizing time taken to do tasks’, as the students following this approach believe that time spent learning is time wasted [12]. The superficial strategies are aimed at mechanical and repetitive learning, following the principle of minimum effort. Students who follow a superficial approach tend to accept information passively, concentrating only on the demands of the exam and not reflecting on the purpose of the information [52]. To obtain academic achievements with the least effort possible, they try to predict what the evaluation criteria are going to be, the type of exam and the concrete demands of each lecturer, focusing on those aspects that provide good results, minimizing effort and personal involvement in the learning process [52,53]. The expectations of success and the academic performance will be higher in those students following the deep approach; while those who follow a superficial approach are motivated by fear of failure [52]. The students who opt for the Deep Strategies, aimed at a meaningful understanding of new knowledge, need to trust in their own possibilities and capabilities to achieve their objectives, using diverse cognitive resources. That is, they are motivated by a strong perceived capacity and a positive consideration of themselves as students [52,54].

Which factors of the Good Practices have been associated with the Deep Strategies? According to our findings, those students who wish to learn by cooperating with their colleagues tend to adopt Deep Strategies. Some previous studies have found that preferences for different levels of interactivity in educational methods gave rise to differences in the approaches to learning and their educational preferences [40]. Those students who used Deep Strategies preferred interactive educational methods, unlike those who scored high in the Superficial Strategy. Discussing the task and the contents with colleagues in order to enrich their point of view was the behavior pattern associated with the deep approach [12]. These results coincide with the research into the positive effects that cooperative learning has on academic, affective and social variables. As for the academic variables, the results of a meta-analysis [55] verified that cooperation is superior to competition and individuality as far as the performance and productivity of all the participants was concerned. The cooperative learning environment is more dynamic, attractive and amusing; while giving the students more responsibility and power over their own learning, increasing their perception of their autonomy and their perceived competence. Cooperative learning improves the quality of the learning strategies, develops deeper strategies for processing information and favors critical and constructive thought [56,57]. The cognitive effectiveness of cooperative learning over the quality of the learning strategies is due, principally, to the fact that the process of discussion that takes place between the students in teamwork situations promotes the discovery and development of cognitive strategies of a higher quality [58].

As for Achievement Strategies, we have observed that they are associated with the need to optimize the time taken to do tasks. Those students who use Achievement Strategies tend to focus on what is important: being organized and managing their time efficiently. They avoid tasks that will not be valued and tend to carry out the demands made concerning work, deadlines and optional subjects [12,50]. We have not found an association between this learning approach and the ‘Cooperative work’ factor. What is more, we have observed a negative relation with the ‘Respect for Different Capacities’ factor, as well as with background and learning styles of colleagues from the IBPEU. As it is an approach that considers good marks to be a priority and thus competing with colleagues is necessary, cooperative work will clearly not be one of their priorities. They will participate in cooperative activities when absolutely necessary to obtain a good mark, but they will not look for any learning benefits from it. The real motivation of this approach involves raising self-esteem and the importance of «me» through success (motive) [59,60], (programing and organizing their time and resources (strategy) in order to achieve higher qualifications [52]. The items that make up the ‘Respect for different capacities’ factor are associated with an empathic and inclusive view of higher education and have very little to do with the competitive and individualistic search for academic achievements.

### Limitations of this Study

This is a transversal study, so causal links cannot be made. Similarly, the sample size means the possibilities of generalizing the results are limited. On the other hand, the evaluation of the variables is solely based on self-reporting. We have chosen to focus on the normally described aspects concerning the students themselves and we have not gathered information concerning their perceptions about the teaching methodology for learning developed in the classroom. Nevertheless, it seems to us that our results point to the importance of making lecturers aware of this model of Students’ Good Practices and training them in those that are more desirable at a university level.

## 5. Conclusions

The results obtained here allow us to assert that there is a relation between students’ behavior patterns (Good Practices) and their different learning strategies. We believe this to be an interesting contribution, since students’ Good Practices can be trained and evaluated and they are closely related to the perspective of learning focused on the student proposed by the European Higher Education Area (EHEA). For this reason, the dimensions of the IBPEU are a promising alternative for evaluating students’ skills aimed at being successful in the context of learning derived from the EHEA. The independence of the different dimensions allows us to easily identify the most relevant aspects of students’ behavior; aspects that can be reinforced through a specific intervention program, adapted to the needs of a particular student or work group. They can easily be applied by lecturers due to their brevity. As they are simple, easy to understand self-reporting surveys, the IBPEU scales allow us to easily access large samples of students, a very desirable circumstance for educational research. Another advantage of this model is that it is based on a model of good practices, for teaching at university level [41], which offers lecturers seven proposals to improve their teaching while also encouraging good practices in the students. As with learning strategies, for the students to work cooperatively, for instance, the lecturer must propose group tasks and use a marking style that favors cooperation over competition. In order to favor good use of time, the lecturer should be very clear on the deadlines for handing in finished tasks and advise students on the effort they should be making to do the said task, etc. 

Finally, if what we want are students who use deep learning strategies, according to our results, these strategies are mainly associated with ‘Cooperative work’, ‘Actively learning’ and less ‘Interaction with lecturers’. One important suggestion for teaching staff is to apply active and participative methodologies in the classroom based on teamwork, in which new material can be shared and discussed, while also learning to manage group processes. Cooperative, collaborative or other forms of group learning are being used more and more in university classrooms in order to teach the students to work in groups, improve performance/learning outcomes and develop fundamental organizational competences for information, communication, conflict management, etc.; all of which are essential for students’ personal and professional growth. The process of discussion and debating ideas that occurs between students in teamwork situations promotes active learning. In addition, cooperative learning makes students responsible for their own learning and gives them greater autonomy and independence from the lecturer [61,62].

## Figures and Tables

**Figure 1 ijerph-17-01849-f001:**
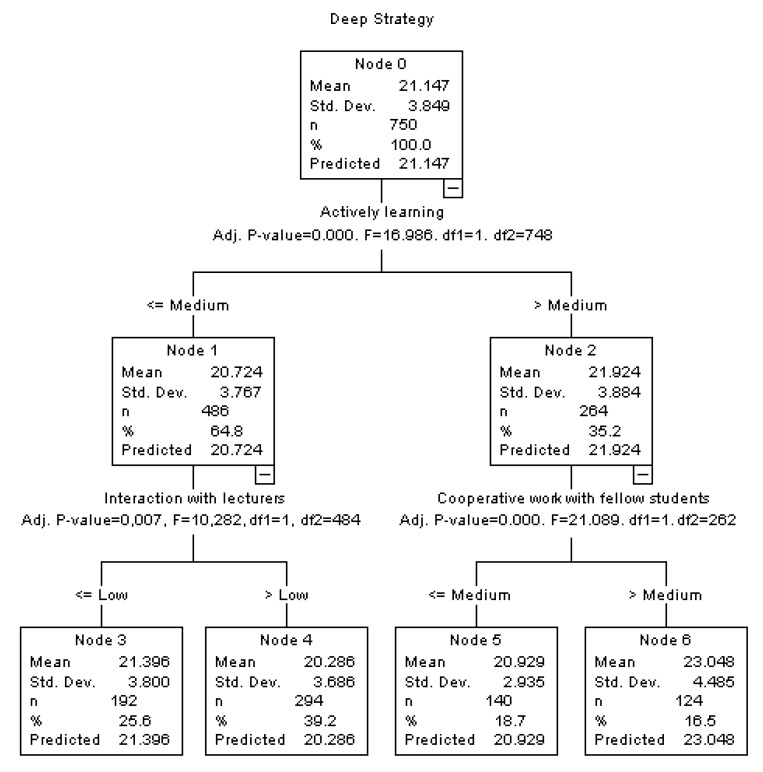
Classification tree for Deep Strategy.

**Figure 2 ijerph-17-01849-f002:**
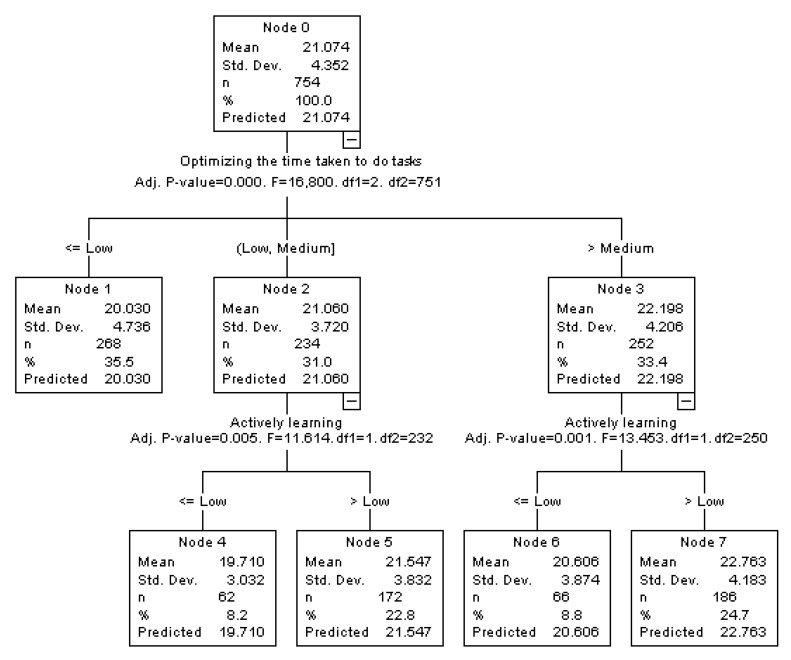
Classification tree for Achievement Strategy.

**Table 1 ijerph-17-01849-t001:** Values of CR, Ω and St.α of the CEPEA and IBPEU scores.

Variables	Composite Reliability	McDonald’s Omega	Standardized Cronbach’s Alpha
Superficial Strategy	0.775	0.777	0.741
Deep Strategy	0.719	0.745	0.753
Achievement Strategy	0.722	0.718	0.711
Interaction with lecturers	0.788	0.788	0.785
Cooperative work with fellow students	0.823	0.821	0.813
Actively learning	0.825	0.820	0.814
Looking for feedback	0.786	0.764	0.761
Optimizing the time taken to do tasks	0.896	0.892	0.891
Mantaining positive expectations	0.790	0.786	0.778
Respecting different capacities, backgrounds and ways of learning	0.886	0.886	0.884
Managing academic challenges and opportunities	0.940	0.940	0.940
Managing personal and social resources	0.929	0.929	0.929

(CEPEA) University students’ Questionnaire to Evaluate Study and Learning Processes and (IBPEU) University Students’ Good Practice Inventory.

**Table 2 ijerph-17-01849-t002:** Estimates of the covariance parameters.

Dependent Variables	Parameter	Estimate	Standard Error	Wald Z	*p*
Deep Strategy	Residues	12.955	0.670	19.313	0.000
	Centers/Faculties	3.353	2.830	1.185	0.236
Achievement Strategy	Residues	17.804	0.919	19.364	0.000
	Centers/Faculties	2.115	1.850	1.143	0.253
Superficial Strategy	Residues	13.717	0.708	19.362	0.000
	Centers/Faculties	0.534	0.529	1.010	0.313

**Table 3 ijerph-17-01849-t003:** Factors associated with learning strategies based on a linear regression analysis.

Dependent Variables	Predictor Variables	B	SE	β	*t*	*p*	*Colinearity*	
							*Tolerance*	*FIV*
Deep Strategy	Constant	18.313	0.88		20.815	0.000		
	Cooperative work	0.70	0.029	0.103	2.391	0.017	0.708	1.411
	Actively learning	0.205	0.041	0.232	5.037	0.000	0.614	1.629
	Interaction with lecturers	−0.120	0.033	−0.157	−3.652	0.000	0.707	1.414
Achievement Strategy	Constant	17.923	0.928		19.320	0.000		
	Optimizing the time taken to do tasks	0.188	0.037	0.322	5.048	0.000	0.917	1.091
	Actively learning	0.149	0.037	0.149	3.989	0.000	0.975	1.025
	Respecting different capacities	−0.089	0.040	−0.141	−2.215	0.027	0.315	3.179
Superficial Strategy	Constant	16.942	0.631		26.865	0.000		
	Interaction with lecturers	0.172	0.027	0.233	6.358	0.000	1	1

B = unstandardized regression coefficient. β = standardized regression coefficient. *t* = obtained *t*-value. p = probability. Dependent variable: emotional instability.

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
