# Peer review of "Good Practices and Learning Strategies of Undergraduate University Students"

_ijerph, 2020, doi:10.3390/ijerph17061849_

Round 1

Reviewer 1 Report

The paper studies Students’ learning strategies and their good practices which are related to their learning  strategies. The author samples 610 students from different science degree courses at University of Extremadura who anonymously completed the University Students’ Good Practice Inventory (IBPEU) and the University students’ Questionnaire to Evaluate Learning Processes (CEPEA). Then the discussion and conclusion are given at last.

The methodology is sound and convincing. The structure is logically coherent and integrity. The result is significant and enlightening to teaching staff in pedagogy. However some tips are suggested to perfect the paper. First it is important to add some relevant literature review in the beginning. Second it is necessary to introduce what is good practice in classroom teaching.  

1. It is important to add some relevant literature review in the beginning.

2. It is necessary to introduce what is good practice in classroom teaching and correspond to the conclusion in the end.

Author Response

Dear  Reviewer 1:

We would like to thank you all for considering your carefully review in order to improve the final manuscript. We are pleased to submit a revised version of our article, which has been substantially improved as a result of your comments and suggestions. We have tried to modify every aspect that has been pointed out (see changes in red). The manuscript also has been checked by a professional translator, native speaker in English. We hope that you will consider this new version acceptable for its publication. We remain at your disposal for any questions you would have

Responses to Review

  1. It is important to add some relevant literature review in the beginning.

A new paragraph with 7 bibliographic references related to the model has been added to the introduction of the article.

  2. It is necessary to introduce what is good practice in classroom teaching and correspond to the conclusion in the end.

We deeply appreciate this suggestion, which allows us to reinforce the usefulness of the Model of Good Practices and highlight the performance of the teacher, from the model proposed by Chickering & Gamson (1987). It is evident that the role of the teacher will be key to create dynamics in the classroom that favor the good practices of the student. We had not realized the importance of this idea. We have introduced an explanatory paragraph in the conclusions section.

Sincerely,

The authors.

Reviewer 2 Report

The paper presents an attempt to present an analysis of  students' behavior patterns in order to predict most productive learning pattern.

The results are interesting and could be useful for professionals in the field of higher education, however, I would like to suggest some improvements.

1) The title could be changed to the more concrete one 

2) Abstract: it is not clear, what deep, achievement and superficial approach means. The main idea is covered in introduction, so it would be better to describe it briefly in Introduction section. 

3) Regarding characteristics of the students, I would also recommend considering differences between bachelor and master students. Master students normally are more independent and better organised.

4) What is lack in this paper is visualisation of the information obtained (graphs or diagrams would improve an impression).

5) For me the connection between strategies and students is still not clear (how did you get what kind of strategy student normally choose) 

6) There are some typos (line 46, page 2: ample (should be sample), line 57, it's better to use another word instead of "furthermore")

Author Response

Dear  Reviewer 2:

We would like to thank you all for considering your carefully review in order to improve the final manuscript. We are pleased to submit a revised version of our article, which has been substantially improved as a result of your comments and suggestions. We have tried to modify every aspect that has been pointed out (see changes in red). The manuscript also has been checked by a professional translator, native speaker in English. We hope that you will consider this new version acceptable for its publication. We remain at your disposal for any questions you would have.

Review:

  • The title could be changed to the more concrete one:

Title: We have introduced in the title a more detailed definition of the participants, specifying that they are undergraduate students.

  • Abstract: it is not clear, what deep, achievement and superficial approach means. The main idea is covered in introduction, so it would be better to describe it briefly in Introduction section. 

Abstract: The three types of strategies definition (superficial, deep and achievement, with an explanatory term for each of them) have been included in the Abstract.

3) Regarding characteristics of the students, I would also recommend considering differences between bachelor and master students. Master students normally are more independent and better organised.

We have added the term “undergraduate” to the title of the article and the inclusion criteria, since only students from the different UEX grades participated. We appreciate the suggestion of the reviewer and consider that as a future line of research, it may be interesting to explore the learning strategies employed by graduate students and compare them with those of undergraduate students.

  • What is lack in this paper is visualisation of the information obtained (graphs or diagrams would improve an impression).

Graphs: We appreciate your suggestion but after thinking about it we have come to the conclusion that the decision trees are a good graphic representation of the results obtained. We understand that adding new graphics would be redundant.

  • For me the connection between strategies and students is still not clear (how did you get what kind of strategy student normally choose).

The Learning Strategies have been evaluated based on the participants' responses to the Questionnaire to Evaluate the Learning and Study Process of University Students (CEPEA), in the three subscales of whichevaluate the learning strategies: Superficial Strategy, Deep Strategy and Achievement Strategy. As explained in the “Instruments” section. The representation of the values obtained by our participants in these subscales can be seen in Tables 2 and 3. The averages and standard deviations of the Deep, Surface and Achievement Strategy subscales have been represented in the decision trees (Figures 1 and two).

Our objective has not been to classify the students in the different strategies, but to look for what dimensions of the good practices of the students are associated with the different types of strategies. In this sense, the direct scores of the IBPEU scales have been modified, based on their percentiles, to establish a categorical scale of three levels (low, medium and high) in those Good Practice subscales that have shown a clear relationship with the Learning strategies in the regression analysis. The CEPEA subscale scores corresponding to the Learning Strategies have not undergone any transformation.

Sincerely,

The authors.

Reviewer 3 Report

The issue raised in this study appears theoretically relevant and is practically important for university educators. It contributes to the discussion regarding ways of positively influencing students' approaches to learning. The results of this study have also potential for maximising teaching designs and learning environment in Higher Education. The authors have used an appropriate methodology, which is clearly explained, and have conducted a very thorough analysis. 

On the whole, this is a well written article where the authors follow a logical development and therefore it reads fairly easily. I shall indicate in dot points below minor corrections to improve the quality of the paper further:

  • L.112 reference [37] should really be [39] which would coincide with the author mentioned, Pinhiero. It's also possible that in that paragraph the sequence of reference numbers is missing a couple. This needs to be checked
  • L115, remove the comma between the words nature and help us, since the subject should not be separated from its verb.
  • L121 "university of Extremadura", the country location should be added for clarity and non-european readers
  • L329 delete "to define" (which may have been left from a previous draft)

Author Response

Dear  Reviewer 3:

We would like to thank you all for considering your carefully review in order to improve the final manuscript. We are pleased to submit a revised version of our article, which has been substantially improved as a result of your comments and suggestions. We have tried to modify every aspect that has been pointed out (see changes in red). The manuscript also has been checked by a professional translator, native speaker in English. We hope that you will consider this new version acceptable for its publication. We remain at your disposal for any questions you would have

Sincerely,

The authors.

Reviewer 4 Report

Interesting and well-written manuscript.  Four learning approaches were considered:  actively learning, interaction with lecturers; cooperative work, and achievement approach.  The authors clearly presented the results. The authors focused on exploring what relationships the students ' chair and manner of interacting with learning contexts can have with the strategies that define their learning. They were able to conclude there definitely is a relation between learning patterns and Learning Strategies as Good Practice.  Connecting each of the four approaches identified were both negatively and positively associated with deep an achievement approaches,  superficial and deep approaches,  deep and optimizing time and respect for different capabilities.  The results were well-defined and explained.

Author Response

Dear  Reviewer 4:

We would like to thank you all for considering your carefully review in order to improve the final manuscript. We are pleased to submit a revised version of our article, which has been substantially improved as a result of your comments and suggestions. We have tried to modify every aspect that has been pointed out (see changes in red). The manuscript also has been checked by a professional translator, native speaker in English. We hope that you will consider this new version acceptable for its publication. We remain at your disposal for any questions you would have.

Sincerely,

The authors.